# Innovativeness as the Key to MSMEs' Performances

Annuridya Rosyidta Pratiwi Octasylva [1,2,*], Lilik Noor Yuliati [3], Hartoyo Hartoyo [1] and Agus W. Soehadi [4]

1   School of Business, IPB University, Bogor 16151, Indonesia; hartoyo@apps.ipb.ac.id
2   Department of Management, Institut Teknologi Indonesia, South Tangerang 15314, Indonesia
3   Department of Family and Consumer Sciences, IPB University, Bogor 16680, Indonesia; lilikny@apps.ipb.ac.id
4   School of Business and Economics, Universitas Prasetiya Mulya, Tangerang 15339, Indonesia; aws@pmbs.ac.id
*   Correspondence: annuridyar@apps.ipb.ac.id or annuridya.rpo@iti.ac.id

**Abstract:** Research on Micro Small Medium Enterprises (MSME) has always been an exciting area of study because of its crucial role; however, it turns out that MSMEs have many problems. The problems, such as human resources and their abilities, are rarely discussed. MSMEs tend to be formed because of a compulsion to do so, whereas companies are created by opportunities; therefore, it is essential to include entrepreneur orientation and dynamic capability elements in making MSME performance models. This study used SEM analysis with 333 respondents spread across six provinces in Indonesia. The results showed that innovativeness, which is part of entrepreneur orientation, is critical in the formation of MSMEs' performances. Moreover, based on the calculation of indirect effects, it revealed that innovativeness through marketing capabilities has a significant influence on the performance of MSMEs.

**Keywords:** performance; dynamic capability; entrepreneur orientation; innovativeness; MSMEs





## 1. Introduction

Research on Micro Small Medium Enterprises (MSME) has become an exciting topic [1]. In addition, research on entrepreneurship is associated with other research fields such as sociology and psychology [2]. Continued development of this research topic is due to its essential role in the economy to overcome the unemployment rate, its contribution to social development and economic growth [3,4], as well as to reduce poverty [5]. The role of MSMEs in overcoming unemployment is seen in the territories of the Organization for Economic Cooperation and Development (OECD) countries. MSMEs represent almost the entire business population, whose distribution equates to approximately 70% of those in employment, and they produce an economic contribution of 50% to 60% of income [6]. Even during the economic recession in 2009, MSMEs still comprised up to 95% of all companies in OECD countries [7]. The number of MSMEs in the world, which accounts for 70% of all businesses globally, makes MSMEs a research subject that deserves attention.

Considering their crucial role, there are some factors that hinder MSMEs' performance, for instance: human resources/social capital/norms [8–10], marketing [8,9,11], low productivity [5], lack of management ability [6,9], low access to technology [10], finance [8–11], bureaucracy, government, policy [12], infrastructure [8], competitiveness [10,11], and low innovation [13,14]. In addition, the source of the problem of the poor performance of MSMEs (MSMEs cannot develop) is the low dynamic ability of MSMEs. Thus, it is not easy to adapt to dynamic environmental conditions.

MSMEs must have a competitive advantage in order to perform sustainably [15]. The competitive advantage can be achieved when the company earns more economic benefits than break-even point competitors, and when it delivers unique advantages with regard to cost, product differentiation, and portfolio advantage over its competitors [16]. This competitive advantage is formed from something unique, difficult to imitate, and is typically called a resource-based view [17]. Entrepreneur orientation is an essential factor

in helping organization performance [18]. The modest (small) organizational structure of MSMEs makes entrepreneur orientation a vital point in forming the dynamic capabilities to maximize business performance [19]. Moreover, the entrepreneur orientation of MSMEs needs to be taken into consideration as it is related to the ability of an entrepreneur to capture available opportunities as a part of their business strategies, and it is key to MSMEs' performance [20].

Apart from entrepreneur orientation, dynamic capability (DC) also plays a critical role in terms of sustainable performance in dynamic environments. Dynamic capability refers to the company's ability to create new forms of competitive advantage. Dynamic capability refers to the ability of company to become flexible, to create, integrate, build, extend, and reconfigure internal and external competencies to cope with rapid/dynamic environmental changes [21,22]. The dynamic ability of a company or organization is the ability to manage the dynamic capabilities in order to survive and compete in a market with rapid/dynamic changes. Dynamic ability refers to a knowledge that can generate value for the company; the knowledge is acquired from both the results of innovation and the transformation of input into output that eventually aims to obtain a sustainable competitive advantage [23].

Even though there has been a great deal of research on this subject, it is still rarely implemented in developing countries, particularly with regard to the relationship between entrepreneur orientation and dynamic capability in terms of MSME performance. Previous studies have reported that entrepreneur orientation (EO) and its dimensions may explain the performance of MSMEs [24]. The main difference in this study is that entrepreneur orientation is created by capturing opportunities. Moreover, MSMEs are created because of a compulsion to do so or because the entrepreneur has no other choice but to do so [25]. Nevertheless, the positive relationship between entrepreneur orientation (EO) and performance is not always significant [26,27] and can be influenced by other factors.

Innovativeness is part of an organization culture where management is open to new ideas, and will implement them to solve organizational problems [28]. The innovations implemented in an organization allow it to accept more innovative ideas. These organizations are believed to be more successful in reacting to environmental changes and have a better market position; therefore, innovation has become the main tool that enables companies to survive. In a highly innovative organization, top-level managers are proactive in seeking creative ideas, but they are also prepared to accept these ideas at their own risk (risk-taking).

Thus, this study aims to explore variables in entrepreneur orientation (proactiveness, innovativeness, and risk-taking) [29] through the dynamic capability (marketing capability, adaptive capability, and absorptive capability) of performance. The study focuses on the organization level and focuses on the configuration of the entrepreneur organizations [30]. This study employs quantitative research using SEM analysis to examine the suitable factor models in forming performance. This research sample includes 333 MSMEs. This study tries to contribute to the entrepreneurial world and the MSME literature by explaining the role of various combinations of factors that determine the company's performance.

This study starts with a literature review to find the research gap and create a hypothesis. Then, it is followed by an explanation of the methods, the power analysis, discussion, and finally the conclusion, followed by suggestions for following researchers can be found at the end of this study.

## 2. Literature Review

### 2.1. Entrepreneur Orientation, Dynamic Capability and Performance

MSMEs, with their characteristics and personal orientation, influence the growth of their companies. The EO refers to a strategic position that reflects how companies implicitly and explicitly choose to compete. In other words, the EO consists of the processes, practices, and decision-making styles of the owner-manager or company involved in entrepreneurial activities [31]. According to Adam, EO includes the processes, habits, and management decision-making styles used in entrepreneurship [32].

The entrepreneur orientation represents policies and practices that provide the basis for the entrepreneurship. The EO roots derive from a body of knowledge on the strategy-making process. EO is considered to aid the process of creating an entrepreneurial strategy that key decision-makers use to implement their business organization's goals, maintain their vision, and create a competitive advantage, ultimately develops performances.

In this study, the EO is divided into three parts, as follows: innovativeness (innovative orientation) refers to a broad concept which notes the tendency of companies to innovate, accept new ideas, encourage experimentation, and support change [29]; proactiveness is an essential attitude in developing a business, as it signifies who moves quickly and responsively [33]; risk-taking refers to the company's willingness to seize opportunities in an uncertain business environment [34].

DC is a derivative of the Resources Based View (RBV) and the company's ability to integrate, build, and reconfigure internal and external competencies to cope with rapid/dynamic environmental changes [35]. The dynamic capability is a particular ability of the company that can utilize resources to create profit. In addition, it can be used to explain how a combination of competencies and resources that are valuable, rare, and difficult to imitate and substitute can be developed, disseminated, and protected. Some abilities can influence the performances of other companies, such as managerial ability, organizational ability, innovation ability, adaptability, dynamic managerial ability, and business process capability. All these abilities are influenced by external factors in their development. Variables used in forming hypotheses in this study are part of the dynamic capability theory, such as adaptive capability, absorptive capability, and dynamic capability [36].

A performance is the result achieved from work that has been implemented; in other words, it is the result of cooperation between members to achieve organizational goals [37]. Sonnentag and Sabine defined performances as processes which result from a person's behavior (in this case, MSME owners) [38]. Performances are behaviors that happen during a variety of activities that indicate "quality", which can occur in various situations [39]. According to Prasanna, the performances of a business can be divided into two parts: survival and development (number of employees, profits, and assets) [40].

### 2.2. Hypothesis Development

The ability to take risks has a significant impact on many businesses, as they learn to adapt, then take advantage of the opportunities available, in order to survive the dynamic conditions [34]. The encouragement of risk plays a role in terms of how it impacts a business' ability to participate. MSMEs that dare to take risks will better acknowledge these conditions, and thus, they have the best attitude to survive.

Proactiveness (Pro) refers to the alertness, readiness, preparedness, and company's capabilities and expectations to create and advance development [29,41]. Thus, it could refer to a responsive person, who is responsible for developing the market.

A business that is filled with proactive people tends to be able to absorb information and knowledge concerning how to grow the business. Proactiveness can impact a person's willingness to adapt to a new environment so that the individual can improve upon their previous abilities. Proactive behavior and initiative are essential determinants of organizational success.

The company's ability to recognize the new value of external knowledge, and then, assimilate and exploit it in its operations or for commercial purposes, can be achieved by having a proactive attitude. This is in line with Basco's research stating that in terms of entrepreneur orientation, proactiveness influences absorption capacity [18]. Proactiveness is a process of adoption, an initiative that influences the environment to gain profit, pursue opportunities, and desire aggressively in order to achieve a satisfactory business performance [42].

Innovativeness (Ino) describes the tendency to engage and support new ideas, novelty, experimentation, and creative processes that can produce new products, services, or technological processes [43]. Innovation among people can encourage improvements in the

management of organizations, processes, and products marketed [44]. When it is achieved, it can improve the business' ability to manage existing or targeted markets. The level of innovation of a person will impact the business' ability to manage the market better. This is indicated by one's ability to optimize one's resources to achieve strategic goals. Innovativeness is needed to make changes in order to improve performances [45].

High adaptability stems from the ability to innovate. In this case, it makes the ideas expressed more realistic and applicable. Biedenbach and Müller explained that innovation is the activity of introducing new things to others, and these new things impact the ability to adapt [46]. Innovation is considered as being key to the sustainability of a company. An innovation made through the development of new products is one way for a company to maintain sustainable growth with respect to their business performances [47]. Innovations implemented continuously by someone to absorb new things improves the ability to manage the company.

Marketing capability (MC) refers to the ability to handle the marketing mix strategies, (i.e., pricing, sales, communication, and product development) [48]. This ability allows the company to establish and implement new strategies to create the company's performances in terms of organizational goals, by responding to changing market conditions. Thus, MC is an essential driver to turn valuable resources into satisfactory company performances. An absorptive capability refers to a company's capacity to deploy resources that have been combined in a process, and how the company can utilize external influences internally in the research and development process [46], where absorptive capacity can be a significant contributor to performance.

An adaptive capability (Adap) is a company's ability to change its understanding of business market expectations by identifying and maintaining key capabilities, resources, and other organizational processes. Biedenbach and Müller said that the adaptive capability is the ability to identify and take advantage of growing market opportunities [46]. This capability can significantly contribute to producing performances that are separate from the absorptive capability [46].

Therefore, based on the abovementioned literature, the hypotheses of the present study are as follows:

**Hypothesis 1 (H1).** *Risk Taking has a significant effect on Adaptive Capability.*

**Hypothesis 2 (H2).** *Proactiveness has a significant effect on Marketing Capability.*

**Hypothesis 3 (H3).** *Proactiveness has a significant effect on Absorptive Capability.*

**Hypothesis 4 (H4).** *Proactiveness has a significant effect on Adaptive Capability.*

**Hypothesis 5 (H5).** *Innovativeness has a significant effect on Marketing Capability.*

**Hypothesis 6 (H6).** *Innovativeness has a significant effect on Absorptive Capability.*

**Hypothesis 7 (H7).** *Innovativeness has a significant effect on Adaptive Capability.*

**Hypothesis 8 (H8).** *Marketing Capability has a significant effect on Performances.*

**Hypothesis 9 (H9).** *An Absorptive Capability has a significant effect on Performances.*

**Hypothesis 10 (H10).** *An Adaptive Capability has a significant effect on Performances.*

The research model can be seen in Figure 1.

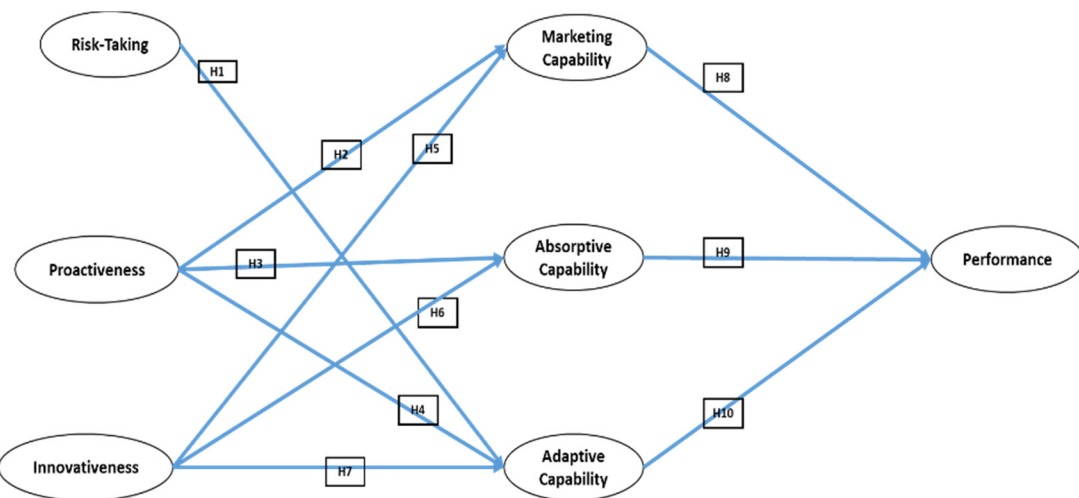

**Figure 1.** Research Model.

## 3. Methods

### 3.1. Data Collection and Sampling

The research design used in this study was a conclusive and quantitative method that utilized the approach of dynamic capability integration models, entrepreneur orientation, and the social capital theory. Then, data was collected via Google Forms (online survey method) between January and April 2021, in 6 provinces in Indonesia, which have the highest number of MSMEs in Indonesia [49].

The research used nonprobability sampling (purpose sampling) whose respondents were taken from MSMEs. The MSMEs are categorized into two types, such as new entrepreneurs (business duration 3–42 months) and established entrepreneurs (business duration > 42 months), following Global Entrepreneur Monitoring (GEM). The number of samples used in this study included 333 MSMEs, which have been included in the minimum sample number, according to Hair [50], and are in-between 5–10 of the indicator. The research used Structural Equation Modeling (SEM) analysis because it can test research hypotheses [32] using software analysis. The SEM model was developed from 2 analyses: the measurement and structural models. The relationship between latent variables and their indicators is validated by the measurement model (validity and reliability), whereas the structural model is tested through the degree of conformity of the data, along with the model (through GOFI) and the significance of the coefficient tested variables (through the *t*-value) [51].

### 3.2. Variable and Measure

The study used five Linkert scales, and the two theories used in this study were the entrepreneur orientation (EO) and the dynamic capability (DC) for the performance formation. Then, the performances variable was formed from 7 indicators, wherein the EO theory was divided into three variables: innovativeness, proactiveness, and risk-taking. The DC theory, on the other hand, was categorized into three variables: marketing capability, absorptive capability, and adaptive capability. Each variable used in this study was taken from previous research as mentioned in Table 1. The result of the reliability and validity tests in this study can be seen in Table 1.

**Table 1.** Variable and Measurement.

| Theory | Variable | Indicator | SLF | CR | Source Adapted from |
|---|---|---|---|---|---|
| Entrepreneur Orientation | Innovativeness | Internal Ideas | 0.62 | | |
| | (Ino) | Modify | 0.68 | | |
| | | Activities | 0.70 | 0.82 | [13,52–54] |
| | | Adjustment | 0.68 | | |
| | | External Adaptation | 0.60 | | |
| | | Market development | 0.67 | | |
| | Proactiveness | Self-awareness | 0.68 | | |
| | (Pro) | Imagination | 0.64 | | |
| | | Intuition | 0.56 | | |
| | | Desire | 0.73 | 0.85 | [55–58] |
| | | Persistent | 0.68 | | |
| | | Other positions | 0.74 | | |
| | | Flexible | 0.69 | | |
| | Risk-Taking | Risk | 0.67 | | |
| | (Risk) | Try | 0.74 | | |
| | | Brave | 0.66 | 0.83 | [34,53,59–62] |
| | | Calm | 0.69 | | |
| | | Cautious | 0.64 | | |
| | | Consistent | 0.65 | | |
| Dynamic Capability | Marketing Capability (MC) | Management | 0.65 | | |
| | | Resources | 0.66 | | |
| | | Marketing | 0.60 | 0.82 | [52,63–66] |
| | | Skills | 0.60 | | |
| | | Ownership of excellence | 0.73 | | |
| | | Added value | 0.68 | | |
| | Absorptive Capability (Absp) | Suggestion | 0.66 | | |
| | | Update | 0.72 | 0.73 | [67–70] |
| | | Information | 0.68 | | |
| | Adaptive Capability (Adap) | Change | 0.77 | | |
| | | Trend | 0.72 | 0.80 | [71–74] |
| | | Situation | 0.73 | | |
| | | Development | 0.61 | | |
| Performances | Performances (Perf) | Sales | 0.66 | | |
| | | Profit | 0.67 | | |
| | | Market Share | 0.75 | | |
| | | Effective | 0.73 | 0.86 | [75–82] |
| | | Competitive | 0.59 | | |
| | | User | 0.69 | | |
| | | Quality of products/services | 0.70 | | |

## 4. Results

Table 2 described the distribution of MSMEs, where most of MSMEs did not have legal entities (sole proprietorship 96.40%), with more than 50% of MSMEs being female (60.96%). Moreover, in terms of education level, the majority of MSME owners were high school graduates (63.06%). In addition, when business duration was reviewed [83], 191 MSMEs (57.36%) were approximately 3 to 42 months old, and 142 MSMEs were established MSMEs, with the majority owners falling within the age ranges for the Z and X generations. The largest number of respondents came from Banten, Indonesia (48.95%).

**Table 2.** Respondent profile.

| Category | | Sum | Percentage |
|---|---|---|---|
| Legal Entity | Sole proprietorship | 321 | 96.40 |
| | CV | 7 | 2.10 |
| | Coorporation | 4 | 1.20 |
| | Ltd., Inc. | 1 | 0.30 |
| Gender | Male | 130 | 39.04 |
| | Female | 203 | 60.96 |
| Education | Elementary School | 9 | 2.70 |
| | Junior High School | 24 | 7.21 |
| | Senior High School | 210 | 63.06 |
| | Bachelor Degree | 84 | 25.23 |
| | Magister Degree | 6 | 1.80 |
| Age (Generation) | Baby Boomer | 24 | 7.21 |
| | X generation | 113 | 33.93 |
| | Y generation (Milenial) | 78 | 23.42 |
| | Z generation | 118 | 35.44 |
| Area | Banten | 163 | 48.95 |
| | Central Java | 65 | 19.52 |
| | West Java | 31 | 9.31 |
| | East Java | 51 | 15.32 |
| | DKI Jakarta | 21 | 6.31 |
| | North Sumatera | 2 | 0.60 |

### 4.1. The Measurement and Structural Models

The validity test revealed that the entire indicator was valid, with the SLF value above 0.5. There were four indicators whose SFL values were less than 0.5; thus, it is excluded from the model. Moreover, the result of the reliability test showed that all variables had a CR value of >0.7, but the absorptive capability variables (0.69), and all AVE, wereabove 0.43. This was considered acceptable, according to Hair's research [51,84]. Thus, all variables were considered valid and reliable.

### 4.2. The Model Fit

The GOF was used to acknowledge the measurements models that functioned to validate the research models. The test results showed the goodness of fit index, such as RMSEA = 0.059 (Good Fit); NFI = 0.94 (Good fit), NNFI = 0.97 (Good fit); PNFI = 0.87; CFI = 0.97, GFI = 0.81. Thus, it can be concluded that this model was good fit. Figure 2 showed the research model with the result of the hypothesis analysis. The tests used in the research hypothesis used the *t*-value to determine the outcome of the hypothesis. Table 3 display the result of structural equation model analysis.

**Table 3.** Result of Structural Equation Model analysis.

| Hypothesis | Effect | Relationship | Coefficient | *t* Value | Result |
|---|---|---|---|---|---|
| H1 | | Risk → Adap | 0.06 | 0.66 | Not support |
| H2 | | Pro → MC | −0.17 | −1.66 | Not support |
| H3 | | Pro → Absp | 0.18 | 1.57 | Not support |
| H4 | | Pro → Adap | 0.13 | 1.23 | Not support |
| H5 | Direct Effect | Ino → MC | 0.95 | 7.73 | Support |
| H6 | | Ino → Absp | 0.49 | 4.09 | Support |
| H7 | | Ino → Adap | 0.63 | 6.01 | Support |
| H8 | | MC → Perf | 0.42 | 4.63 | Support |
| H9 | | Absp → Perf | 0.056 | 0.75 | Not support |
| H10 | | Adap → Perf | 0.22 | 2.58 | Support |
| | Indirect Effect | Ino → Perf | 0.56 | 6.94 | Support |
| | | Risk → Perf | 0.01 | 0.64 | Not support |
| | | Pro → Perf | −0.03 | −0.57 | Not support |

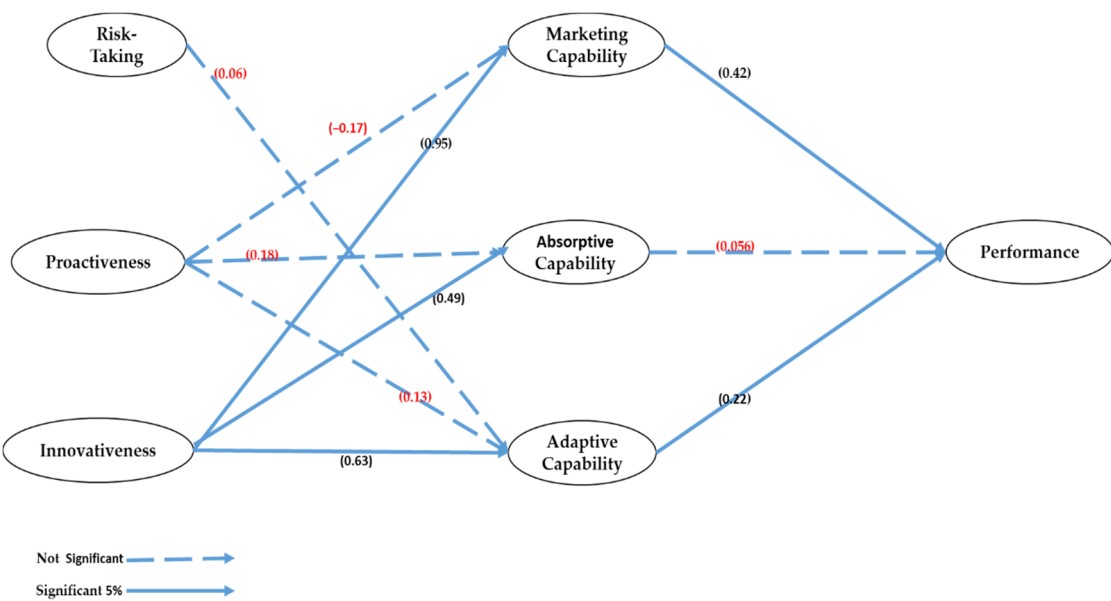

**Figure 2.** Result Model.

## 5. Discussion

This study aimed to examine factors that influence the MSMEs' performances and the role of the dynamic capability in ensuring the sustainability performances. The risk-taking variable in this study was unable to support the adaptation ability of MSMEs (H1). Thus, it was different from the results of previous studies [85]. The companies that increase entrepreneur orientation as a type of strategy will obtain better results if MSMEs understand market conditions and have the courage to take risks. The MSMEs' competitiveness is significantly related to the level of risk-taking. The result showed that MSMEs with reasonable levels of risk-taking are more likely to perform better than those that have high or low levels of risk-taking.

Risk-taking that involves the tendency to commit a significant proportion of resources, such as investment in high-risk projects that promise considerable returns, as well as a pre-disposition for big lending in order to meet the uncertainty condition, make it possible for MSMEs to improve performance levels. These findings provide a boost for the resource-based theory, by demonstrating the important role of risk-taking as a strategy that leads to a competitive advantage and satisfactory MSME performance [86]; however, the results of this study show that the risk-taking variable does not significantly increase the ability of MSMEs to adapt.

The results showed that proactiveness does not support the MSMEs' capabilities H2–H4. This contradicts previous research [87] stating that proactiveness influences the dynamic capability. This shows that although MSMEs are proactive in various ways, their proactiveness does not improve their marketing, adaptability, and absorption capabilities. The low level of marketing is also influenced by the difficulties that MSMEs face in terms of adapting, as it considers the need for investment [88]. This finding is interesting, since proactiveness does not significantly influence marketing capability, absorptive capability, and adaptive capability, even though it has an essential role in business.

Moreover, the dynamic capability theory approach shows that the marketing capability and adaptive capability influence performance, as in Hypothesis 8 and Hypothesis 10. This happens because the dynamic capability theory asserts that resources are not enough to create innovation. The study's findings suggest that entrepreneurs should equip themselves with the ability to capture what is happening in the ecosystem and combine it with the internal resources to innovate [51].

The results of this study are in line with previous studies in that the marketing capability could form a competitive advantage and eventually improve the performances of MSMEs [48,89].

In addition, the marketing capability directly influences performances the most compared with other factors, such as adaptive capability and absorptive capability. It happens, as in this dynamic environment, it is vital to have marketing capabilities so that the products marketed can touch consumers directly. Moreover, the absorptive capability also influences performance. This study gives different results from earlier studies [90], stating that absorptive capability is an essential/significant factor for building performances. MSMEs can continue increasing their capacity to achieve satisfactory performances with the ability to absorb various information, knowledge, and technology in an ever-changing environment.

The most exciting finding in this study is that innovativeness influences performances, both directly and indirectly. The hypothesis test shows that for H5–H7, innovativeness has a significant effect on dynamic capability (marketing, absorptive, and adaptive capabilities). Innovativeness is the tendency to introduce new things by developing new products, services, processes, technologies, and models [91]. A willingness to innovate and to produce something new in order to achieve a competitive advantage will make MSMEs more dynamic, and eventually achieve better performances. This shows that the higher the innovativeness of a MSME, the more likely it is that the adaptive capability of a MSME will increase. The results of this study are quite relevant to the results of a study from Eshima, which states that the improvement of adaptive capability is due to increasing entrepreneurial activity, since companies increase their entrepreneur orientation as a mechanism strategy to utilize their better understanding of market conditions [85]. This study shows that if MSMEs have a high level of innovation, they will tend to experience a greater ability to absorb knowledge and adapt to a dynamic environment, as well as obtaining stronger marketing capabilities.

In terms of marketing capability, the higher the orientation to innovate (innovativeness), the more the marketing capability of MSMEs will improve. Innovation is essential for organizational sustainability, as companies need to be innovative to survive in a highly competitive environment. In addition, innovation allows companies to respond to opportunities and competitive threats, make fast and customer-oriented decisions, and develop new products to meet market needs and the expectations of customers, all of which helps companies develop better than their competitors. The willingness to innovate, accompanied by the ability to innovate, are recognized as some of the determining factors for organizations to survive and succeed. Innovation is vital in organizations as the success of new products is an engine of growth that strongly affects sales, profits, and competitiveness. A company's competitive advantage depends heavily on its relationships with external organizations rather than its internal capability. The results of this study are in line with the previous research stating that entrepreneur orientation influences marketing capability and can help form a competitive advantage [48,92]. MSMEs realize that innovation is an essential factor in improving the performances of MSMEs, so the willingness to innovate in MSMEs is also high.

The sixth hypothesis test is on the influence of innovativeness variables on absorptive capability. The influence test shows that the innovativeness variable significantly influences absorptive capability with a path coefficient of 0.49 and a *t*-test of 4.09. The result shows that the higher the innovativeness of MSME, the more the absorptive capability of MSMEs will increase. Based on the result of the study, only innovation indirectly builds performances. This is in line with the results of previous studies stating that companies that have high innovativeness will also have good performances [93,94]; therefore, it proves that both directly and indirectly, the key to a satisfactory performance is the innovativeness of MSMEs.

## 6. Managerial Implications

Managerial implications were obtained from discussions and observations in the field, and open interviews with several MSMEs and experts were supported by literature studies. High innovativeness means an entrepreneurial orientation with the desire to continue to innovate that will improve the performances of MSMEs, which includes performances in terms of sales, profit, market share, effectiveness, competition, number of customers, and quality.

The results of this study are significant for MSMEs and stakeholders to pay more attention to factors that can improve the performances of MSMEs by focusing on innovativeness and marketing capabilities. This research is important because it identified an essential factor that helps shape the performances of MSMEs. When we know the key factors that improve performance, we can improve the welfare of the community by improving the welfare of MSMEs, as most businesses in various countries are micro-constrained.

In general, the empirical findings of this study provide practical implications for stakeholders of the structural model, in terms of the driving factors of the formation of dynamic capability that can serve as a mediator in the formation of MSMEs performances. More specifically, MSMEs can use the results of this study to identify factors that affect business performances which can be applied in strategy development.

## 7. Limitations and Future Research

The research conducted has several limitations. The first limitation is that the structural model is only limited to entrepreneur orientation and dynamic capability factors by analyzing MSME performances models in general, within the scope of research in Indonesia. The second limitation is that this research does not cover various existing conditions; for instance, it does not conduct MSME research into various business sizes (small/medium) using multi group analysis.

Our research results have limitations as this research stems from an entrepreneur orientation and dynamic capability approach. Suggestions for further research could include other various factors such as social capital, business ecosystem, and entrepreneur intention. Thus, it can fill the gap in this research.

## 8. Conclusions

This study shows that the majority of MSME owners are female, and in terms of education level, they are primarily high school graduates. The characteristics of the MSME profile form the performance model of MSMEs. The study results revealed that risk-taking was not proven to have a significant direct effect on increasing the ability of MSMEs to adapt. MSMEs with a reasonable level of risk-taking are more likely to have a better performance than those with low or high risk-taking.

The role of proactiveness does not significantly affect marketing capability, absorptive capability, and adaptive capability. The dynamic capability approach to performance reveals that marketing capability and adaptive capability will directly form the arrangement of MSMEs. MSMEs need to improve their abilities to absorb various information, knowledge, and technology in a dynamic environment to develop MSME performance properly.

The most exciting finding in this study is that innovativeness has a significant effect on the performance of MSMEs. The capability of MSMEs to innovate and produce something new to achieve a competitive advantage can make them move dynamically to improve their marketing capabilities, adaptive capabilities and performance. This capability is a strategic mechanism for understanding current market conditions in a better way. Innovation also enables MSMEs to respond to competitive opportunities and threats, make quick and customer-oriented decisions, and develop new products to meet market needs and customer expectations. The role of innovation also significantly influences the absorptive capabilities of MSMEs. With innovativeness, MSMEs will always strive to create new ideas, which requires information and knowledge from internal and external companies.

**Author Contributions:** Conceptualization, A.R.P.O., L.N.Y., H.H. and A.W.S.; Data curation, A.R.P.O.; Formal analysis, A.R.P.O.; Funding acquisition, A.R.P.O.; Investigation, A.R.P.O.; Methodology, A.R.P.O.; Project administration, A.R.P.O.; Resources, A.R.P.O.; Software, A.R.P.O.; Supervision, L.N.Y., H.H. and A.W.S.; Validation L.N.Y., H.H. and A.W.S.; Writing—original draft, A.R.P.O. All authors have read and agreed to the published version of the manuscript.

**Funding:** The APC is partially funded by IPB University and Institut Teknologi Indonesia.

**Institutional Review Board Statement:** Not applicable.

**Informed Consent Statement:** Not applicable.

**Data Availability Statement:** Not applicable.

**Conflicts of Interest:** The authors declare no conflict of interest.

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
