# Peer review of "Innovativeness as the Key to MSMEs’ Performances"

_sustainability, doi:10.3390/su14116429_

Round 1
Reviewer 1 Report
The authors attempt to contribute to the understanding of innovativeness as the key to MSMEs Performance. The paper endeavors to make a contribution by examining the mediating effects of marketing capability on the relationship between innovativeness and firm performance.
I welcome your attempt to tackle the issue of the role of innovativeness in explaining MSMEs performance.
The introductory segment was very informative and provided the objectives of the paper, identified the theoretical contrasts and gaps within the existing literature, and explained the contributions of the manuscript well.
The hypotheses development section provides a good combination of literature on entrepreneur orientation, dynamic capability, and performance.
In the methods section, part of the problem, for me at least, is your sampling frame. What is the meaning of ‘online media in 6 provinces of Indonesia’ in your empirical setting? Please explain this more specifically.
In the conclusion, there was not much description. A bit more elaboration on the results, managerial implications, and future studies would be helpful.
Overall, this paper is quite interesting, but I think it needs more development in the method section and in the conclusion.
Good luck with this project.
Author Response
Response to Reviewer 1 Comments
Point 1: In the methods section, part of the problem, for me at least, is your sampling frame. What is the meaning of ‘online media in 6 provinces of Indonesia’ in your empirical setting? Please explain this more specifically.
Response 1:
I use a sampling method with non-probability sampling. This data collection was online using Google Forms. This study was conducted in 6 provinces with the largest number of MSMEs, according to Indonesia's national statistics agency. These six provinces represent 63.17% of Indonesia's MSMEs, representing the picture of MSMEs in Indonesia.
Point 2: In the conclusion, there was not much description. A bit more elaboration on the results, managerial implications, and future studies would be helpful
Response 2:
The conclusion section has been corrected by covering all 10 of the hypothesis testing. And at the end of the discussion, a section has been added on managerial implications and future studies.
Point 3: Overall, this paper is quite interesting, but I think it needs more development in the method section and in the conclusion
Response 3:
Improvements in the research methods section have been done. And the conclusion has already been developed.
Reviewer 2 Report
Dear authors,
Thank you for the paper. Itwas pleaseant to read, however there are a few things you need to solve, especially with the English language:
- When you first use acronims in the Abstract such as "MSMEs", please state what it means.
- To Literature Review section, please cite more recent papers.
- You state to the hypothesis H2, H3 and H4 Proactive has a significant ....proactive what? Please review the english
- In table 2 what is Persentase? did you mean Percentage? Jumlah = SUM???
- The Conclusion part need to be extended.
Author Response
Response to Reviewer 2 Comments
Point 1: 1. When you first use acronims in the Abstract such as "MSMEs", please state what it means.
Response 1:
MSMEs is the abbreviation for Micro Small Medium Enterprises. I already corrected it on the Abstract.
Point 2: To the Literature Review section, please cite more recent papers
Response 2:
The literature review has been updated using the recent papers.
Point 3: You state to the hypothesis H2, H3 and H4 Proactive has a
significant ....proactive what? Please review the english
Response 3:
What 'Proactive' mean is the state or condition of being proactive by MSMEs. Therefore, I revise proactive to become the proactiveness of MSMEs.
Point 4: In table 2 what is Persentase? did you mean Percentage?
Jumlah = SUM???
Response 4:
Sorry for the missing type I have made.
'Persentase' is in Indonesian, which means percentage, while' Jumlah' means a sum.
Point 5: The Conclusion part needs to be extended.
Response 5:
The conclusions were extended based on input from reviewers, including a summary of the ten results of hypothesis testing, managerial implications, and further study in order to these research topics.
Round 2
Reviewer 2 Report
Dear authors,
Thank you for reviewing and improving your paper.
I think it can be published in present form.